# DermX: a Dermatological Diagnosis Explainability Dataset

**Raluca Jalaboi**[1, 2]**, Mauricio Orbes-Arteaga**[2]**, Dan Richter Jørgensen**[2]**, Ionela Manole**[2]**, Oana Ionescu Bozdog**[2]**, Andrei Chiriac**[2]**, Ole Winther**[1, 3, 4]**, Alfiia Galimzianova**[2]

[1]Section for Cognitive Systems, Technical University of Denmark
[2]Omhu, Denmark
[3]Bioinformatics Centre, Department of Biology, University of Copenhagen
[4]Centre for Genomic Medicine, Rigshospitalet, Copenhagen University Hospital
{rjal, olwi}@dtu.dk, {raluca, mauricio.orbes, dan, ionela.manole,
oana.bozdog, andrei.chiriac, alfiia}@omhu.com

## Abstract

In this paper, we introduce DermX: a novel dermatological diagnosis and explanations dataset annotated by eight board-certified dermatologists. To date, public datasets for dermatological applications have been limited to diagnosis and lesion segmentation, while validation of dermatological explainability has been limited to visual inspection. As such, this work is a first release of a dataset providing gold standard explanations for dermatological diagnosis to enable a quantitative evaluation of ConvNet explainability. DermX consists of 525 images sourced from two public datasets, DermNetNZ and SD-260, spanning six of the most prevalent skin conditions. Each image was enriched with diagnoses and diagnosis explanations by three dermatologists. Supporting explanations were collected as 15 non-localisable characteristics, 16 localisable characteristics, and 23 additional terms. Dermatologists manually segmented localisable characteristic and described them with additional terms. We showcase a possible use of our dataset by benchmarking the explainability of two ConvNet architectures, ResNet-50 and EfficientNet-B4, trained on an internal skin lesion dataset and tested on DermX. ConvNet visualisations are obtained through gradient-weighted class-activation map (Grad-CAM), a commonly used model visualisation technique. Our analysis reveals EfficientNet-B4 as the most explainable between the two. Thus, we prove that DermX can be used to objectively benchmark the explainability power of dermatological diagnosis models. The dataset is available at `https://github.com/ralucaj/dermx`.

## 1 Introduction

Convolutional neural models (ConvNets), the current state-of-the-art method for image analysis, are often criticised for being opaque in their decision mechanisms [1]. However, explainability is a crucial component in the adoption of machine learning systems in high-stakes applications, such as medical diagnosis. Dermatology in particular would highly benefit from automation, given the low diagnostic accuracy of general practitioners [2] and the scarcity of specialists [3, 4]. Deep learning methods to diagnose skin conditions exist [5–8], but their adoption by the medical system has been slow, partially due to their lack of explainability [9, 1, 10].

Different research groups proposed various explainability methods [11–13], but their use has been limited to visual inspection of the outputs to evaluate model performance. Such an approach is subjective and difficult to scale. Lesion segmentation masks offered by high quality dermatology

Submitted to the 35th Conference on Neural Information Processing Systems (NeurIPS 2021) Track on Datasets and Benchmarks. Do not distribute.

Table 1: Distribution of images over the public datasets. Initially, 100 images were randomly selected for each class, apart from viral warts and vitiligo where only 78 and 88 images were available. Some images were discarded during labelling, giving rise to the count shown below.

| | Acne | Actinic keratosis | Psoriasis | Seborrhoeic dermatitis | Viral warts | Vitiligo |
|---|---|---|---|---|---|---|
| DermNetNZ | 52 | 48 | 46 | 12 | 47 | 75 |
| SD-260 | 47 | 43 | 51 | 66 | 27 | 11 |

datasets [14] can partially serve as a basis for objective measurement, although they were not collected to explain the diagnosis. However, this shortcoming becomes critical in diseases such as actinic keratosis, where the surrounding area is just as important for the diagnosis as the lesion itself [8].

We introduce DermX, a novel dermatological diagnosis explainability dataset that addresses the limitations of existing datasets by collecting dermatologist explanations for six skin diseases: acne, actinic keratosis, psoriasis, seborrhoeic dermatitis, viral warts, and vitiligo. Each image is diagnosed by three dermatologists and tagged with supporting characteristics [15] and their localisation.

To demonstrate the intended use of DermX, we benchmark two models trained to diagnose dermatological conditions. We employ gradient-weighted class-activation maps (Grad-CAM) [13], a deep learning visualisation technique commonly used to generate explanations, on ResNet-50 [16] and EfficientNet-B4 [17]. Then, we test how their explanations compare to the dermatologist maps.

The contributions of this paper are twofold:

1. We release a novel dermatological diagnosis explainability dataset with annotations from multiple expert raters;

2. We benchmark the explainability of two popular model architectures against a gold standard explainability dataset.

## 2 Dataset

DermX consists of 525 images of acne, actinic keratosis, psoriasis, seborrhoeic dermatitis, viral warts, or vitiligo patients. Eight board-certified dermatologists, with between 4 and 12 years of clinical experience, labelled the images with diagnoses and explanations supporting their diagnoses, in the form of both global tags and characteristic segmentations. The images were randomly selected from DermNetNZ [18] and SD-260 [19], and are available under the Creative Commons licence. Permission to use the data in this project was granted in writing by the owners of both datasets. The distribution of diseases is described in Table 1.

Our work involved several steps. First, we performed several experiments to define the target diseases and the nature of the explanations. Second, we defined the diagnosis and explanation ontology, as illustrated in Figure 1. Then, the labellers were allowed a short period of time to get accustomed to the annotation protocol and the labelling tool by evaluating images from an internal dataset. Finally, DermX images were selected and sent to the dermatologists for labelling.

### 2.1 Preliminary Investigation

Nine diseases were initially investigated: psoriasis, rosacea, vitiligo, seborrhoeic dermatitis, pityriasis rosea, viral warts, actinic keratosis, acne, and impetigo. These diseases were chosen based on prevalence [20] and the expectation that they could be diagnosed only from images [21]. Dermatologists were asked to diagnose and explain their diagnosis in free-text for around 100 images. This step led to both the exclusion of rosacea, impetigo, and pityriasis rosea from future experiments due to the difficulty in diagnosing them in the absence of an anamnesis, and to the introduction of a structured ontology for the diagnosis explanations to avoid manual processing of typos and synonyms.

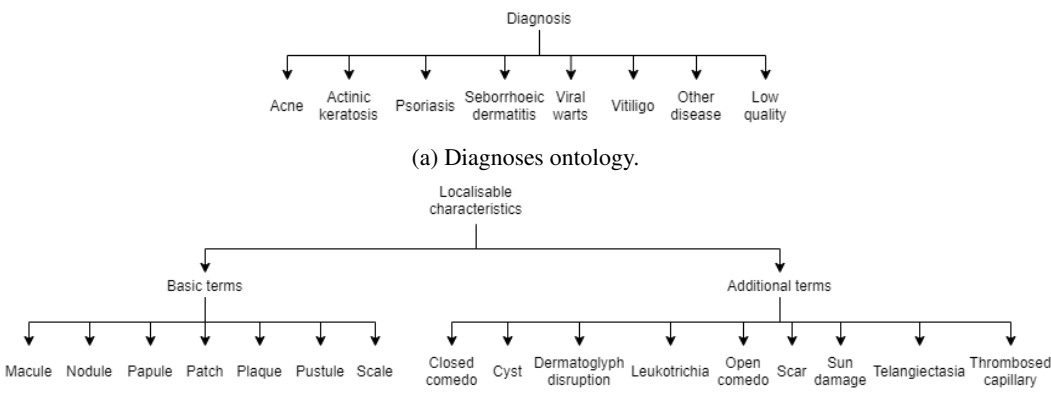

(a) Diagnoses ontology.

(b) Localisable characteristics ontology.

Figure 1: Ontology of the two main types of labels. The list of diagnoses (a) includes the six diseases and two discard options. Discard options could be chosen when images displayed another disease or when images were of low quality. Localisable characteristics (b) were tailored to the six diseases using medical resources [15, 21], and with the help of two senior dermatologists.

## 2.2 Diagnosis and Explanation Ontology

Preliminary investigations highlighted the importance of having a consistent explanation ontology. After analysing free-text explanations, they were formalised as an extended list of skin lesion characteristics [15]. The characteristics set was selected to sufficiently explain the six target diseases [21]. With the help of two senior dermatologists, several other relevant characteristics were added.

The resulting set of characteristics was split into non-localisable characteristics (e.g. age or sex), localisable characteristics (e.g. plaque or open comedo), and additional descriptive terms (e.g. red or well-circumscribed), according to the International League of Dermatological Societies (ILDS) classification [15]. To match state-of-the-art ConvNet explainability methods, we focus on diagnoses and localisable characteristics. Figure 1 illustrates the final DermX ontology, while more information about the other two types of labels is available in Appendix Figure 1.

## 2.3 Annotation Protocol

Dermatologists were first asked to diagnose the image, and then tag it with characteristics that explain their diagnosis. If the dermatologists were unable to evaluate the image due to poor quality, or if the image depicted a different disease than the target conditions, they had the option to discard it.

Dermatologists could then select diagnosis-relevant non-localisable characteristics as global image tags. Afterwards, they could select and localise localisable characteristics. Dermatologists were instructed to highlight all relevant areas for each characteristic, and were only allowed to include irrelevant areas if separating them from the characteristic was too time consuming or difficult. In other words, they were instructed to favour sensitivity over specificity. Finally, basic terms (as defined in Figure 1b) could be enriched with additional descriptive terms when required for the diagnosis explanation. Once all tags and characteristics were added, the image could be marked as complete.

After the ontology and annotation protocol were defined, all dermatologists underwent two rounds of on-boarding in Darwin, a browser-based labelling tool [22] (Appendix Figure 2).

## 2.4 Dataset Analysis

Once all evaluations were finished, we analysed the data focusing on dermatologist performance with regards to the gold standard diagnosis and their inter-rater agreement on both diagnoses and supporting characteristics. Figure 2 illustrates an image and its three annotations.

A total of 566 images were evaluated by eight dermatologists. To better understand the data distribution, we tagged each image with a skin tone approximation: light, medium, and dark, equivalent to Fitzpatrick skin tones [23] I-II, III-IV, and V-VI, respectively. As any post-hoc meta-data creation, this

| Image | Dermatologist 1 | Dermatologist 2 | Dermatologist 3 |
|---|---|---|---|
| 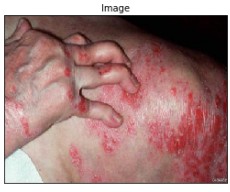 | 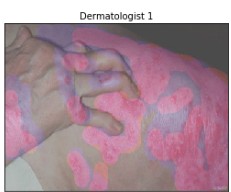 | 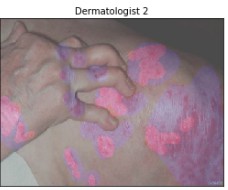 | 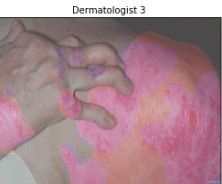 |

Figure 2: Example of a DermX image and its labels from three dermatologists. The blue overlay illustrates the plaque segmentations, while the orange overlay shows the scale segmentations. Pink shows the overlap between the two characteristics. While all dermatologists agree in some areas, there are clear disagreements as to which areas contain a certain characteristic.

labelling task is subject to several sources of error, including lighting conditions, missing information about the patient, and high inter-rater variance for the Fitzpatrick scale [24]. The distribution is skewed towards lighter skin tones, with 368 images, i.e. 65% of the dataset, depicting them. Medium skin tones were illustrated in 182 images, i.e. 32% of data, while darker skin tones only appeared in 16 images, i.e. 3% of the time. A similar analysis, with similar drawbacks, has been performed for the age distribution of patients. Young patients, described as approximately below 30, are depicted in 108 images, i.e. 19% of the dataset. A similar amount of images, 147 or 26% of the data illustrates elderly people, defined as people over 60. The remaining 311 images, i.e. 55% of DermX, showcase adult patients.

From 1698 unique evaluations on 566 images, 411 evaluations were either tagged as other disease or as too low quality to evaluate. These 411 evaluations were removed from the dataset, leading to some images having fewer than three evaluations. Two evaluations tagged an image with multiple diagnoses, and were disregarded from the analysis. Images where all evaluations were discarded were also removed from the dataset. In the rest of the paper, we will only focus on the remaining 1285 evaluations associated with 525 images.

The diagnostic accuracy of dermatologists with regards to the gold standard varies between 0.92 to 0.99. Seborrhoeic dermatitis is the most difficult disease to diagnose, while vitiligo is the easiest. Pair-wise F1 scores for the inter-rater agreement lies between 0.86 and 1.0 (Table 2).

Inter-rater agreement on characteristics (Table 3a) varies significantly more, partially due to the lower number of selections per class. Most basic terms display the highest levels of agreement, with F1 scores between 0.65 and 0.88. The two low performing basic terms, macule and nodule, have low selection rates. Several additional terms as defined in Figure 1b, such as open and closed comedones, display levels of agreement similar to the basic terms.

Outlining characteristics is a more difficult task, as also confirmed by the low inter-rater F1 scores (also known as Dice score when computed for the positive class, see Table 3b). The lower F1 values can also be explained by the annotation protocol specification to prioritise sensitivity over specificity. In terms of sensitivity, we notice the same trend as in the binary agreement: dermatologists tend to agree more on the basic terms. Agreement differences stem from the difficulty in outlining some of these characteristics. For example, comedones cover smaller areas, and dermatologists differed in their approach to outlining them.

Overall, the contrast between high agreement on diagnoses and low agreement on supporting characteristics illustrates how different experts perceive explanations in different ways. Although they generally agree on the diagnosis, dermatologists focus on different characteristics to explain their decision. To properly evaluate a model's explanations, we must therefore consider the opinions of multiple experts.

## 3 Explainability Benchmark for Two Architectures

Using the DermX dataset, we evaluate the explainability of ConvNets trained for skin lesion diagnosis by applying Grad-CAM on two models, ResNet-50 and EfficientNet-B4, and comparing the results to

Table 2: Diagnostic performance (a) and inter-rater agreement (b) on DermX. Dermatologists have high agreement with both the gold standard label and with each other. Seborrhoeic dermatitis stands out as a difficult disease to diagnose, while vitiligo, viral warts and acne appear to be easier.

(a) Dermatologist diagnosis performance with regards to the gold standard (mean±std).

|                       | F1              | Sensitivity     | Specificity     |
|-----------------------|-----------------|-----------------|-----------------|
| Acne                  | $0.98 \pm 0.01$ | $0.99 \pm 0.01$ | $0.99 \pm 0.01$ |
| Actinic keratosis     | $0.94 \pm 0.05$ | $0.92 \pm 0.08$ | $0.99 \pm 0.01$ |
| Psoriasis             | $0.92 \pm 0.03$ | $0.98 \pm 0.02$ | $0.96 \pm 0.02$ |
| Seborrhoeic dermatitis| $0.87 \pm 0.07$ | $0.81 \pm 0.11$ | $0.99 \pm 0.01$ |
| Viral warts           | $0.98 \pm 0.02$ | $0.96 \pm 0.03$ | $1.00 \pm 0.00$ |
| Vitiligo              | $0.99 \pm 0.01$ | $0.98 \pm 0.02$ | $1.00 \pm 0.00$ |

(b) Dermatologist inter-rater agreement on diagnosis (mean±std).

|                       | F1              | Sensitivity     | Specificity     |
|-----------------------|-----------------|-----------------|-----------------|
| Acne                  | $0.95 \pm 0.19$ | $0.95 \pm 0.19$ | $1.00 \pm 0.01$ |
| Actinic keratosis     | $0.90 \pm 0.19$ | $0.91 \pm 0.20$ | $0.99 \pm 0.02$ |
| Psoriasis             | $0.94 \pm 0.07$ | $0.95 \pm 0.09$ | $0.99 \pm 0.02$ |
| Seborrhoeic dermatitis| $0.90 \pm 0.10$ | $0.92 \pm 0.13$ | $0.99 \pm 0.02$ |
| Viral warts           | $0.99 \pm 0.03$ | $0.99 \pm 0.04$ | $1.00 \pm 0.01$ |
| Vitiligo              | $0.98 \pm 0.03$ | $0.98 \pm 0.05$ | $1.00 \pm 0.01$ |

the DermX explanation maps. We selected Grad-CAM for generating the models' attention maps due to its high prevalence in the medical image analysis literature [8, 25]. The Keras [26] implementation of these experiments is available at `https://github.com/leoilab/dermx-experiments`.

## 3.1 Experimental Setup

Both architectures were pre-trained on ImageNet and fine-tuned on 3214 images of the six target skin conditions from an internal clinical dataset. Images from this dataset and their associated diagnoses were obtained during face-to-face consultations with a dermatologist. All patients included in this dataset gave their consent for both research and commercial use of their images. Each model was trained five times, and the results presented are the mean over all trained models.

A hyper-parameter search was run on an 80/20 training/validation split for the internal dataset. We investigated data augmentation parameters (rotation, shear, zoom, brightness), learning rates, and the number of layers to be fine-tuned. Once the optimal hyper-parameter setup was found (Appendix Table 2), the two architectures were trained on the entire internal dataset defined in Table 4. The validation F1 score on the internal dataset was 0.73 for ResNet-50 and 0.79 for EfficientNet-B4. Finally, the models were tested on the 525 DermX images. All experiments were performed on AWS EU (Ireland) instances, summing up to two GPU weeks (NVIDIA V100).

## 3.2 Results

The expected impact of the data distribution shift was made obvious by the model diagnostic performance. Diagnostic accuracy with respect to the gold standard is $0.34 \pm 0.03$, and $0.42 \pm 0.09$ for ResNet-50 and EfficientNet-B4, respectively (Table 5). Both methods represent a significant improvement over the chance accuracy of 0.17. Vitiligo is predicted with both the lowest sensitivity as well as F1 score for both models, while the highest-ranked diagnosis class for both models was acne. As seen in Table 5, EfficientNet-B4 outperformed the ResNet-50 on four out of six diseases, with a difference of 13.5 points in average for F1 score.

We evaluate the explainability of the two ConvNets by comparing their attention maps to the characteristic segmentations. The union of all characteristics segmented by a dermatologist for an image was also compared to the attention map, as a way to check whether the models take into account the entire area selected by dermatologists as important to their decision. To quantify the

Table 3: An inter-rater analysis for supporting characteristics (a) shows significant variation in their selection and agreement rates. Characteristics commonly considered important for diagnosing one of the diseases (e.g. comedones, plaques) have higher agreement rates, while uncommon characteristics (e.g. leukotrichia, telangiectasia) display low selection and agreement rates. Overlap measures (b) show similar differences between raters. Due to the focus on outlining sensitivity at the expense of specificity, most characteristics have a low F1 score. Sensitivity values are high in characteristics that occupy larger areas and that often display well-circumscribed borders (e.g. plaque, scale), but tend to be lower in smaller characteristics (e.g. comedones, pustules).

(a) Dermatologist inter-rater agreement for the presence or absence of characteristics (mean±std).

|  | F1 | Sensitivity | Specificity | Evaluations | Images |
|---|---|---|---|---|---|
| **Basic terms** |  |  |  |  |  |
| Macule | $0.13 \pm 0.24$ | $0.17 \pm 0.31$ | $0.93 \pm 0.10$ | 110 | 93 |
| Nodule | $0.07 \pm 0.22$ | $0.08 \pm 0.26$ | $0.97 \pm 0.05$ | 47 | 44 |
| Papule | $0.65 \pm 0.15$ | $0.69 \pm 0.20$ | $0.86 \pm 0.10$ | 385 | 213 |
| Patch | $0.72 \pm 0.17$ | $0.77 \pm 0.22$ | $0.91 \pm 0.10$ | 335 | 185 |
| Plaque | $0.78 \pm 0.11$ | $0.80 \pm 0.16$ | $0.84 \pm 0.11$ | 592 | 306 |
| Pustule | $0.69 \pm 0.29$ | $0.72 \pm 0.32$ | $0.97 \pm 0.03$ | 161 | 80 |
| Scale | $0.88 \pm 0.09$ | $0.89 \pm 0.12$ | $0.92 \pm 0.09$ | 550 | 257 |
| **Additional terms** |  |  |  |  |  |
| Closed comedo | $0.52 \pm 0.27$ | $0.61 \pm 0.35$ | $0.96 \pm 0.05$ | 108 | 63 |
| Cyst | $0.06 \pm 0.22$ | $0.06 \pm 0.23$ | $0.99 \pm 0.02$ | 16 | 14 |
| Leukotrichia | $0.18 \pm 0.38$ | $0.18 \pm 0.38$ | $1.00 \pm 0.01$ | 12 | 8 |
| Open comedo | $0.65 \pm 0.30$ | $0.71 \pm 0.34$ | $0.97 \pm 0.05$ | 132 | 73 |
| Scar | $0.45 \pm 0.29$ | $0.54 \pm 0.38$ | $0.95 \pm 0.06$ | 112 | 74 |
| Sun damage | $0.46 \pm 0.39$ | $0.49 \pm 0.43$ | $0.97 \pm 0.04$ | 101 | 63 |
| Telangiectasia | $0.08 \pm 0.25$ | $0.09 \pm 0.27$ | $0.99 \pm 0.02$ | 13 | 10 |
| Thrombosed capillaries | $0.31 \pm 0.40$ | $0.35 \pm 0.45$ | $0.97 \pm 0.05$ | 67 | 38 |

(b) Dermatologist inter-rater localisation agreement for localisable characteristics (mean±std).

|  | F1 | Sensitivity | Specificity |
|---|---|---|---|
| **Basic terms** |  |  |  |
| Macule | $0.04 \pm 0.12$ | $0.08 \pm 0.20$ | $0.95 \pm 0.13$ |
| Nodule | $0.03 \pm 0.15$ | $0.06 \pm 0.22$ | $0.98 \pm 0.04$ |
| Papule | $0.20 \pm 0.28$ | $0.33 \pm 0.36$ | $0.96 \pm 0.10$ |
| Patch | $0.45 \pm 0.40$ | $0.59 \pm 0.39$ | $0.93 \pm 0.12$ |
| Plaque | $0.48 \pm 0.39$ | $0.62 \pm 0.37$ | $0.93 \pm 0.12$ |
| Pustule | $0.24 \pm 0.23$ | $0.38 \pm 0.33$ | $0.99 \pm 0.03$ |
| Scale | $0.48 \pm 0.32$ | $0.60 \pm 0.33$ | $0.94 \pm 0.10$ |
| **Additional terms** |  |  |  |
| Closed comedo | $0.08 \pm 0.17$ | $0.24 \pm 0.36$ | $0.93 \pm 0.15$ |
| Cyst | $0.04 \pm 0.13$ | $0.08 \pm 0.18$ | $1.00 \pm 0.01$ |
| Dermatoglyph-disruption | $0.33 \pm 0.41$ | $0.48 \pm 0.42$ | $0.98 \pm 0.04$ |
| Leukotrichia | $0.31 \pm 0.33$ | $0.45 \pm 0.38$ | $0.96 \pm 0.06$ |
| Open comedo | $0.14 \pm 0.19$ | $0.29 \pm 0.33$ | $0.93 \pm 0.15$ |
| Scar | $0.12 \pm 0.23$ | $0.26 \pm 0.36$ | $0.91 \pm 0.14$ |
| Sun damage | $0.35 \pm 0.43$ | $0.51 \pm 0.45$ | $0.75 \pm 0.28$ |
| Telangiectasia | $0.06 \pm 0.16$ | $0.13 \pm 0.25$ | $0.97 \pm 0.05$ |
| Thrombosed capillaries | $0.21 \pm 0.30$ | $0.36 \pm 0.38$ | $0.99 \pm 0.02$ |

similarity between the attention maps and the expert-generated maps, we compute the F1 score, sensitivity and specificity following their fuzzy implementation defined in Crum et al. [27] (Appendix Table 3).

Table 4: Data used for training and testing the methods, split by disease. An internal clinical dataset was employed for training the models, while DermX was used for testing.

|  | Acne | Actinic keratosis | Psoriasis | Seborrhoeic dermatitis | Viral warts | Vitiligo |
|---|---|---|---|---|---|---|
| Training | 1177 | 165 | 975 | 113 | 606 | 178 |
| DermX | 99 | 91 | 97 | 78 | 74 | 86 |

Table 5: Model diagnostic performance with regards to the gold standard, aggregated over five models. ResNet-50 (a) is out-performed on four out of six diseases by EfficientNet-B4 (b). The training data impact can be seen in the high scores for acne and low scores for vitiligo for both models.

(a) ResNet-50 diagnostic performance with regards to the gold standard (mean±std).

|  | F1 | Sensitivity | Specificity |
|---|---|---|---|
| Acne | $0.53 \pm 0.11$ | $0.43 \pm 0.14$ | $0.96 \pm 0.02$ |
| Actinic keratosis | $0.32 \pm 0.11$ | $0.23 \pm 0.12$ | $0.97 \pm 0.02$ |
| Psoriasis | $0.44 \pm 0.04$ | $0.78 \pm 0.20$ | $0.60 \pm 0.18$ |
| Seborrhoeic dermatitis | $0.39 \pm 0.19$ | $0.41 \pm 0.28$ | $0.92 \pm 0.08$ |
| Viral warts | $0.15 \pm 0.06$ | $0.14 \pm 0.07$ | $0.86 \pm 0.04$ |
| Vitiligo | $0.04 \pm 0.02$ | $0.03 \pm 0.02$ | $0.90 \pm 0.05$ |

(b) EfficientNet-B4 diagnostic performance with regards to the gold standard (mean±std).

|  | F1 | Sensitivity | Specificity |
|---|---|---|---|
| Acne | $0.65 \pm 0.32$ | $0.62 \pm 0.33$ | $0.97 \pm 0.02$ |
| Actinic keratosis | $0.55 \pm 0.11$ | $0.45 \pm 0.15$ | $0.96 \pm 0.02$ |
| Psoriasis | $0.57 \pm 0.12$ | $0.90 \pm 0.09$ | $0.67 \pm 0.23$ |
| Seborrheic dermatitis | $0.45 \pm 0.22$ | $0.41 \pm 0.22$ | $0.95 \pm 0.03$ |
| Viral warts | $0.07 \pm 0.04$ | $0.06 \pm 0.04$ | $0.86 \pm 0.04$ |
| Vitiligo | $0.00 \pm 0.01$ | $0.00 \pm 0.01$ | $0.90 \pm 0.03$ |

Similar to the diagnostic performance, the explainability of EfficientNet-B4 is higher than that of ResNet-50 in terms of both F1 score and sensitivity. However, ResNet-50 outperforms EfficientNet-B4 in terms of specificity on most characteristics and on the union of all characteristics (Table 6). These observations are also apparent upon visual inspection of the dermatologists segmentations created by dermatologists and the Grad-CAM visualisations in Figure 3. Much like dermatologists, both models have higher sensitivity scores for basic terms, albeit at a smaller difference. Within additional terms, cyst, scar, and sun damage all reach sensitivity levels similar to basic terms. This may be due to lower selection rates, as is the case for cyst, or because of the larger areas covered by scar and sun damage in images.

## 4   Discussion and Conclusion

Our experiments showcase the intended use of DermX: as an explainability benchmark for dermatological diagnosis ConvNets. By comparing the model explanations to those of the experts not only can we identify the most promising research directions, but also learn about strategies to improve the existing models. For example, if models under consideration systematically miss certain characteristics (i.e. express near-zero sensitivity by never selecting the same areas as the dermatologists), one solution is to ensure that training data represents the characteristic well enough by including both positive and negative samples. Another possible outcome is that models consistently highlight different areas than humans (i.e. express low specificity by including areas deemed irrelevant by the dermatologists). In this case, ensuring the models are not learning irrelevant characteristics might be done by appropriate training data augmentation. Alternatively, if domain experts confirm that the areas highlighted are relevant for the diagnosis, this knowledge might be used to better educate humans, similar to the actinic keratosis seminar held by Tschandl et al. [8].

Table 6: Explainability of ResNet-50 (a) and EfficientNet-B4 (b) as similarity measures between dermatologists-segmented supporting characteristics and model activation maps. For each image, the union of all dermatologist characteristic maps was also compared against the activation maps. All activation maps were computed with regards to the gold standard diagnosis using Grad-CAM.

(a) Explainability of ResNet-50 model (mean±std).

|  | F1 | Sensitivity | Specificity |
|---|---|---|---|
| **Basic terms** | | | |
| Macule | $0.07 \pm 0.02$ | $0.15 \pm 0.02$ | $0.88 \pm 0.02$ |
| Nodule | $0.05 \pm 0.01$ | $0.19 \pm 0.04$ | $0.88 \pm 0.02$ |
| Papule | $0.06 \pm 0.01$ | $0.17 \pm 0.02$ | $0.88 \pm 0.02$ |
| Patch | $0.13 \pm 0.01$ | $0.12 \pm 0.01$ | $0.89 \pm 0.03$ |
| Plaque | $0.18 \pm 0.05$ | $0.19 \pm 0.04$ | $0.89 \pm 0.01$ |
| Pustule | $0.02 \pm 0.01$ | $0.21 \pm 0.08$ | $0.87 \pm 0.02$ |
| Scale | $0.16 \pm 0.05$ | $0.21 \pm 0.05$ | $0.88 \pm 0.01$ |
| **Additional terms** | | | |
| Closed comedo | $0.07 \pm 0.01$ | $0.15 \pm 0.03$ | $0.87 \pm 0.03$ |
| Cyst | $0.03 \pm 0.01$ | $0.21 \pm 0.05$ | $0.86 \pm 0.03$ |
| Dermatoglyph disruption | $0.06 \pm 0.06$ | $0.09 \pm 0.12$ | $0.90 \pm 0.03$ |
| Leukotrichia | $0.11 \pm 0.02$ | $0.14 \pm 0.03$ | $0.90 \pm 0.01$ |
| Open comedo | $0.08 \pm 0.01$ | $0.15 \pm 0.03$ | $0.87 \pm 0.03$ |
| Scar | $0.13 \pm 0.04$ | $0.16 \pm 0.04$ | $0.88 \pm 0.02$ |
| Sun damage | $0.22 \pm 0.04$ | $0.14 \pm 0.03$ | $0.92 \pm 0.06$ |
| Telangiectasia | $0.10 \pm 0.02$ | $0.16 \pm 0.06$ | $0.90 \pm 0.04$ |
| Thrombosed capillaries | $0.03 \pm 0.03$ | $0.10 \pm 0.16$ | $0.91 \pm 0.03$ |
| Union | $0.17 \pm 0.01$ | $0.16 \pm 0.01$ | $0.90 \pm 0.00$ |

(b) Explainability of EfficientNet-B4 model (mean±std).

|  | F1 | Sensitivity | Specificity |
|---|---|---|---|
| **Basic terms** | | | |
| Macule | $0.09 \pm 0.03$ | $0.27 \pm 0.07$ | $0.80 \pm 0.03$ |
| Nodule | $0.03 \pm 0.01$ | $0.20 \pm 0.08$ | $0.82 \pm 0.03$ |
| Papule | $0.07 \pm 0.01$ | $0.23 \pm 0.07$ | $0.80 \pm 0.03$ |
| Patch | $0.21 \pm 0.04$ | $0.28 \pm 0.06$ | $0.80 \pm 0.03$ |
| Plaque | $0.29 \pm 0.01$ | $0.38 \pm 0.03$ | $0.81 \pm 0.04$ |
| Pustule | $0.02 \pm 0.01$ | $0.28 \pm 0.14$ | $0.82 \pm 0.05$ |
| Scale | $0.26 \pm 0.01$ | $0.44 \pm 0.03$ | $0.80 \pm 0.04$ |
| **Additional terms** | | | |
| Closed comedo | $0.08 \pm 0.03$ | $0.21 \pm 0.10$ | $0.83 \pm 0.04$ |
| Cyst | $0.02 \pm 0.01$ | $0.20 \pm 0.14$ | $0.85 \pm 0.03$ |
| Dermatoglyph disruption | $0.05 \pm 0.02$ | $0.09 \pm 0.05$ | $0.79 \pm 0.06$ |
| Leukotrichia | $0.07 \pm 0.03$ | $0.14 \pm 0.10$ | $0.82 \pm 0.04$ |
| Open comedo | $0.08 \pm 0.04$ | $0.19 \pm 0.10$ | $0.83 \pm 0.04$ |
| Scar | $0.16 \pm 0.09$ | $0.25 \pm 0.13$ | $0.82 \pm 0.03$ |
| Sun damage | $0.42 \pm 0.05$ | $0.31 \pm 0.05$ | $0.90 \pm 0.02$ |
| Telangiectasia | $0.14 \pm 0.01$ | $0.35 \pm 0.04$ | $0.79 \pm 0.04$ |
| Thrombosed capillaries | $0.01 \pm 0.01$ | $0.07 \pm 0.02$ | $0.80 \pm 0.05$ |
| Union | $0.25 \pm 0.02$ | $0.29 \pm 0.04$ | $0.82 \pm 0.03$ |

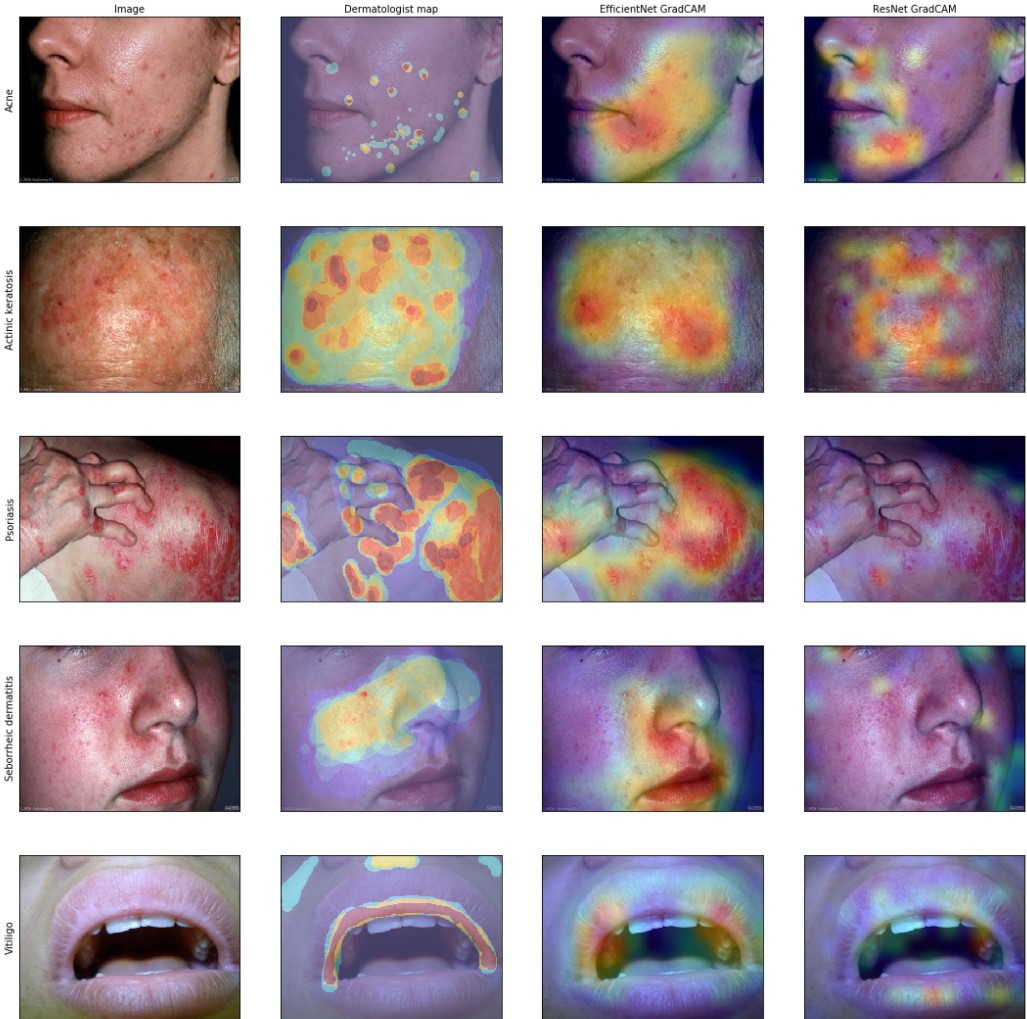

Figure 3: Examples of explanation for images where both models correctly predicted the gold standard diagnosis. From left to right: the original image, the union of all characteristics selected by all dermatologists labelling the image, an EfficientNet-B4 Grad-CAM visualisation, and a ResNet-50 Grad-CAM visualisation. In all cases, the EfficientNet-B4 visualisation is closer to the dermatologist map than the ResNet-50 visualisation. ResNet-50 appears to be more specific, focusing on smaller, more noticeable lesions. More examples can be found in Appendix Figures 4, 5, and 6.

Our benchmarking results demonstrate that there is still a considerable gap among explanations provided by the models trained for this task and the expert dermatologists. For example, the highest sensitivity achieved for a characteristic by a model on the benchmark is $0.44 \pm 0.03$ for scale by EfficientNet-B4, which is still significantly below the expert agreement of $0.60 \pm 0.33$. Building models that can reach expert level, both in terms of the diagnostic performance and the diagnostic reasoning, would require incorporating such expert annotations in the training process. One solution could be using characteristics maps to guide the attention of a model towards the clinically relevant areas in an image. However, collection of such data is a challenging and laborious task, requiring multiple highly trained dermatologists to meticulously segment and tag the data with a rich set of characteristics. From a more practical point of view, we can still draw conclusions on how explainable each model is, even with the low performance observed for both models. DermX can also serve as an external validation dataset for diagnostic tools in general – an important validation aspect of all healthcare-oriented diagnostic tools [28].

The first release of DermX presented in this work has several limitations. First, only a small number of conditions was selected, which, although highly prevalent [20], are not representative of the whole variety of dermatological diseases. One risk associated with this selection is that future explainable models may focus on this smaller set, at the expense of other, more dangerous conditions. Second, expert annotations were limited to up to three dermatological evaluations per image. Diagnostic reasoning is not a simple task, and is subject to inter-rater variability as seen in our analysis in Section 2.4. Increasing the number of the dermatologists per image will help make the measurements more robust. Moreover, the distribution of skin tones in the dataset is skewed towards lighter skin. Although the annotation process was subject to various sources of error, e.g. illumination issues, missing patient information, and labeller experience, the data further highlights the well known low representation of people of colour in publicly available datasets [29]. Finally, in terms of the characteristics chosen, the labelling dermatologists could not select the absence of a characteristic as an important factor in their diagnosis decision.

In the future, we aim to continuously expand the dataset with more data points to enable training of diagnostic models along with learning the supportive characteristics. The dataset will be enriched with more conditions and more dermatologists to make the next DermX releases more comprehensive and objective. We will also expand our labelling protocol by including characteristic negation, and thus expanding the explainability from only supporting characteristics to counterfactual reasoning. In terms of ethical and representation concerns, we aim to select more images illustrating darker skin tones. This action is subject to the availability of such images in published skin lesion datasets.

To conclude, we introduce DermX, the first dermatological dataset created for diagnosis explainability. We expect it to serve as a benchmark to meaningfully improve the performance of the ConvNets built for dermatological diagnosis, and as a possible basis for explainable diagnosis models.

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

 # A  Appendix

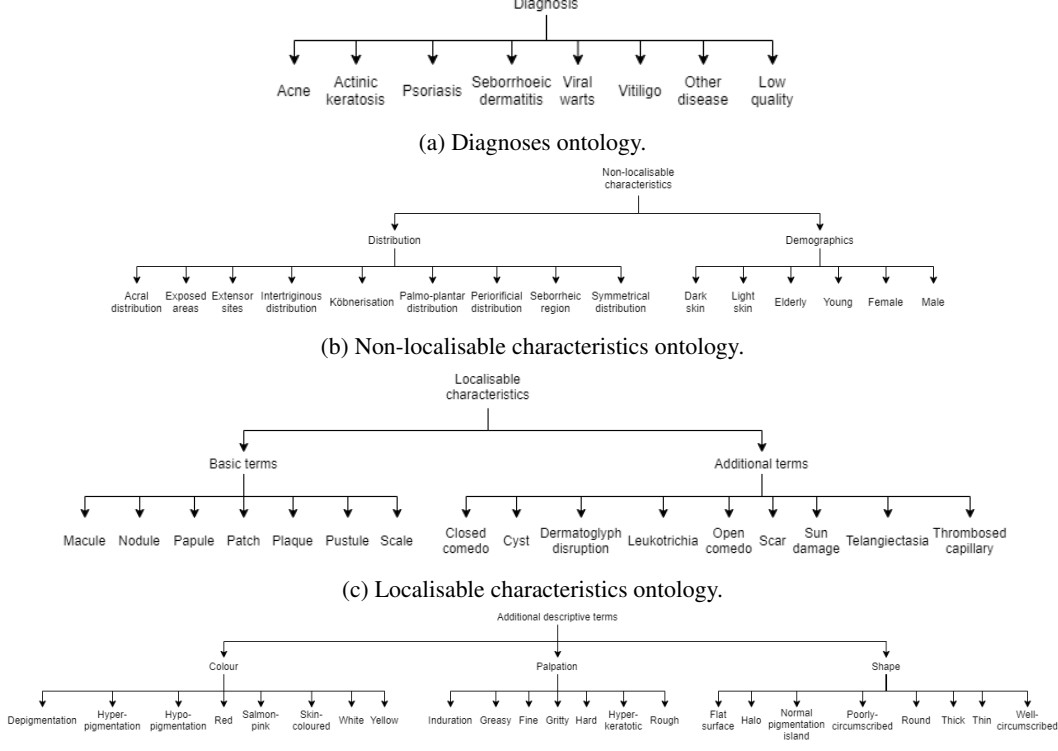

(a) Diagnoses ontology.

(b) Non-localisable characteristics ontology.

(c) Localisable characteristics ontology.

(d) Additional descriptive terms for localisable characteristics.

Figure 1: Ontology of the four types of labels. The list of diagnoses (1a) includes the six diseases and two discard options for images that either displayed another disease or were of low quality. Non-localisable characteristics (1b) were added to the ILDS classification as global image tags after being flagged as relevant by our senior dermatologists. Localisable characteristics (1c) and additional descriptive terms (1d) were tailored for the six diseases from medical resources [15, 21], and with the help of two senior dermatologists.

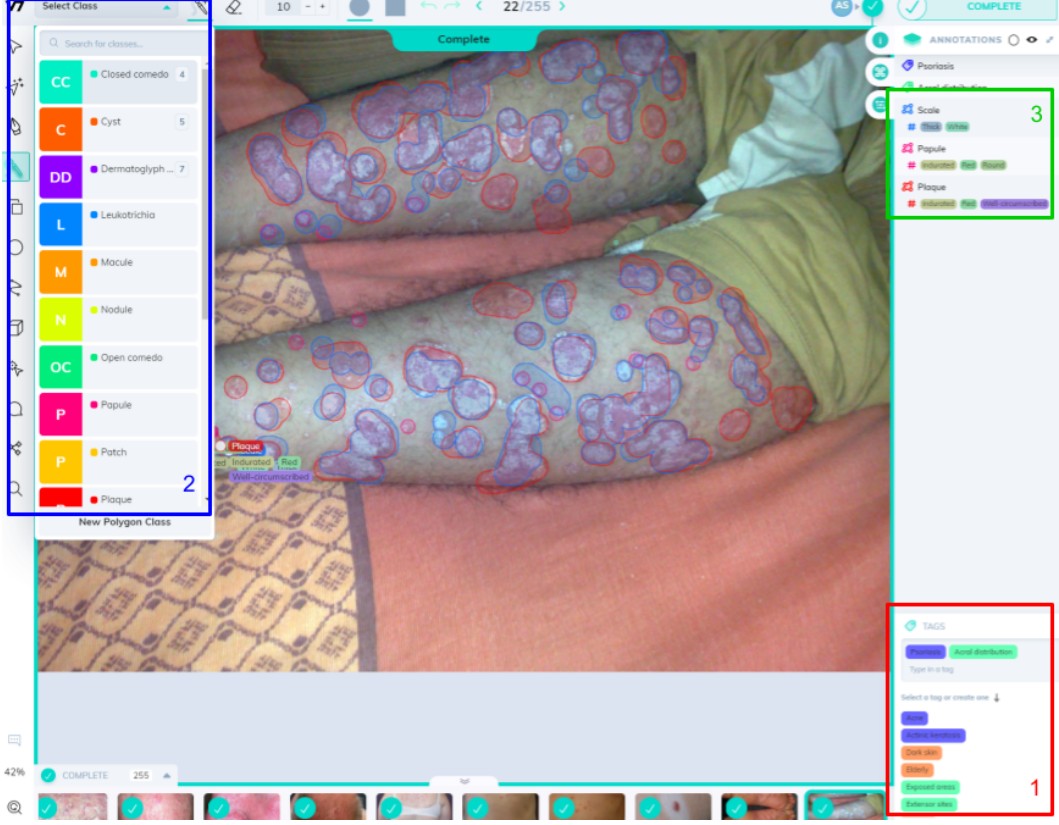

Figure 2: Labelling tool interface, exemplified for a psoriasis case from the SD-260 dataset. In the global tag search box (area 1, bottom right), dermatologists can select the disease, relevant demographics information, and lesion distribution. The brush selection menu (area 2, top left) allows them to select and mark localisable characteristics on the image. The full annotation menu (area 3, top right) is used to select of additional descriptive terms for the localised basic terms.

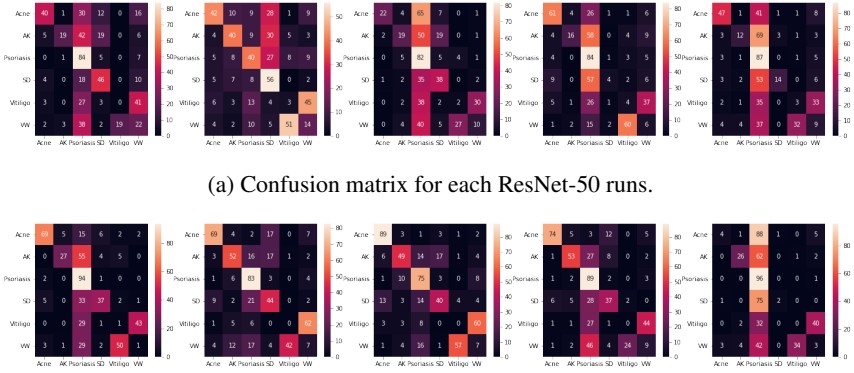

(a) Confusion matrix for each ResNet-50 runs.

(b) Confusion matrix for each EfficentNet-B4 runs.

Figure 3: Confusion matrix for all models trained. Both ResNet-50 (a) and EfficentNet-B4 (b) show a bias towards predicting psoriasis, and predict vitiligo in very few cases.

Table 1: Dermatologist inter-rater agreement for the presence or absence of characteristics, including the number of evaluations (evals) and the number of images where they were identified.

| | F1 | Sensitivity | Specificity | Evals | Images |
|---|---|---|---|---|---|
| **Non-localisable characteristics** | | | | | |
| **Demographics** | | | | | |
| Elderly | $0.50 \pm 0.29$ | $0.58 \pm 0.37$ | $0.93 \pm 0.09$ | 186 | 117 |
| Young | $0.29 \pm 0.29$ | $0.36 \pm 0.39$ | $0.90 \pm 0.13$ | 168 | 123 |
| Female | $0.14 \pm 0.23$ | $0.18 \pm 0.33$ | $0.94 \pm 0.10$ | 84 | 67 |
| Male | $0.14 \pm 0.22$ | $0.20 \pm 0.35$ | $0.91 \pm 0.13$ | 140 | 113 |
| Dark skin | $0.00 \pm 0.00$ | $0.00 \pm 0.00$ | $0.97 \pm 0.05$ | 30 | 30 |
| Light skin | $0.10 \pm 0.22$ | $0.14 \pm 0.31$ | $0.82 \pm 0.27$ | 222 | 193 |
| **Distribution** | | | | | |
| Acral distribution | $0.33 \pm 0.29$ | $0.38 \pm 0.38$ | $0.92 \pm 0.08$ | 149 | 100 |
| Exposed areas | $0.47 \pm 0.33$ | $0.54 \pm 0.38$ | $0.89 \pm 0.12$ | 255 | 172 |
| Extensor sites | $0.28 \pm 0.32$ | $0.31 \pm 0.38$ | $0.95 \pm 0.06$ | 85 | 59 |
| Intertriginous | $0.00 \pm 0.00$ | $0.00 \pm 0.00$ | $0.99 \pm 0.01$ | 9 | 9 |
| Köbnerization | $0.05 \pm 0.20$ | $0.06 \pm 0.23$ | $0.98 \pm 0.02$ | 19 | 17 |
| Palmo-plantar | $0.31 \pm 0.33$ | $0.36 \pm 0.41$ | $0.97 \pm 0.04$ | 74 | 52 |
| Periorificial | $0.16 \pm 0.35$ | $0.16 \pm 0.36$ | $0.99 \pm 0.02$ | 24 | 18 |
| Seborrhoeic region | $0.66 \pm 0.24$ | $0.74 \pm 0.30$ | $0.92 \pm 0.10$ | 287 | 160 |
| Symmetrical | $0.21 \pm 0.24$ | $0.26 \pm 0.33$ | $0.93 \pm 0.07$ | 105 | 85 |
| **Localisable characteristics** | | | | | |
| **Basic terms** | | | | | |
| Macule | $0.13 \pm 0.24$ | $0.17 \pm 0.31$ | $0.93 \pm 0.10$ | 110 | 93 |
| Nodule | $0.07 \pm 0.22$ | $0.08 \pm 0.26$ | $0.97 \pm 0.05$ | 47 | 44 |
| Papule | $0.65 \pm 0.15$ | $0.69 \pm 0.20$ | $0.86 \pm 0.10$ | 385 | 213 |
| Patch | $0.72 \pm 0.17$ | $0.77 \pm 0.22$ | $0.91 \pm 0.10$ | 335 | 185 |
| Plaque | $0.78 \pm 0.11$ | $0.80 \pm 0.16$ | $0.84 \pm 0.11$ | 592 | 306 |
| Pustule | $0.69 \pm 0.29$ | $0.72 \pm 0.32$ | $0.97 \pm 0.03$ | 161 | 80 |
| Scale | $0.88 \pm 0.09$ | $0.89 \pm 0.12$ | $0.92 \pm 0.09$ | 550 | 257 |
| **Additional terms** | | | | | |
| Closed comedo | $0.52 \pm 0.27$ | $0.61 \pm 0.35$ | $0.96 \pm 0.05$ | 108 | 63 |
| Cyst | $0.06 \pm 0.22$ | $0.06 \pm 0.23$ | $0.99 \pm 0.02$ | 16 | 14 |
| Dermatoglyph disruption | $0.32 \pm 0.37$ | $0.33 \pm 0.39$ | $0.97 \pm 0.04$ | 86 | 50 |
| Leukotrichia | $0.18 \pm 0.38$ | $0.18 \pm 0.38$ | $1.00 \pm 0.01$ | 12 | 8 |
| Open comedo | $0.65 \pm 0.30$ | $0.71 \pm 0.34$ | $0.97 \pm 0.05$ | 132 | 73 |
| Scar | $0.45 \pm 0.29$ | $0.54 \pm 0.38$ | $0.95 \pm 0.06$ | 112 | 74 |
| Sun damage | $0.46 \pm 0.39$ | $0.49 \pm 0.43$ | $0.97 \pm 0.04$ | 101 | 63 |
| Telangiectasia | $0.08 \pm 0.25$ | $0.09 \pm 0.27$ | $0.99 \pm 0.02$ | 13 | 10 |
| Thrombosed capillary | $0.31 \pm 0.40$ | $0.35 \pm 0.45$ | $0.97 \pm 0.05$ | 67 | 38 |

Table 2: Optimal hyper-parameter setup and other training parameters for ResNet-50 and EfficientNet-B4, as identified after a hyper-parameter search.

| | ResNet-50 | EfficientNet-B4 |
|---|---|---|
| Rotation | 20 | 20 |
| Shear | 0 | 0.5 |
| Zoom | 0.25 | 0.5 |
| Brightness | 0.25-1 | 0.5-1 |
| Learning rate | 0.01 | 0.001 |
| Optimiser | Adam | Adam |
| Training epochs | 30 | 15 |
| Image size | $300 \times 400$ | $300 \times 400$ |
| Weighted classes | On | On |

Table 3: Similarity metrics used for comparison of models attention maps ($\mathcal{A}$) and dermatologists characteristics segmentations ($\mathcal{S}$).

| Similarity metric | Formula |
|---|---|
| F1 score | $\dfrac{2\sum_{p\in pixels}\min(\mathcal{A}_p,\mathcal{S}_p)}{\sum_{p\in pixels}(\mathcal{A}_p)+\sum_{p\in pixels}(\mathcal{S}_p)}$ |
| Sensitivity | $\dfrac{\sum_{p\in pixels}\min(\mathcal{A}_p,\mathcal{S}_p)}{\sum_{p\in pixels}(\mathcal{S}_p)}$ |
| Specificity | $\dfrac{\sum_{p\in pixels}\min(1-\mathcal{A}_p,1-\mathcal{S}_p)}{\sum_{p\in pixels}(1-\mathcal{S}_p)}$ |

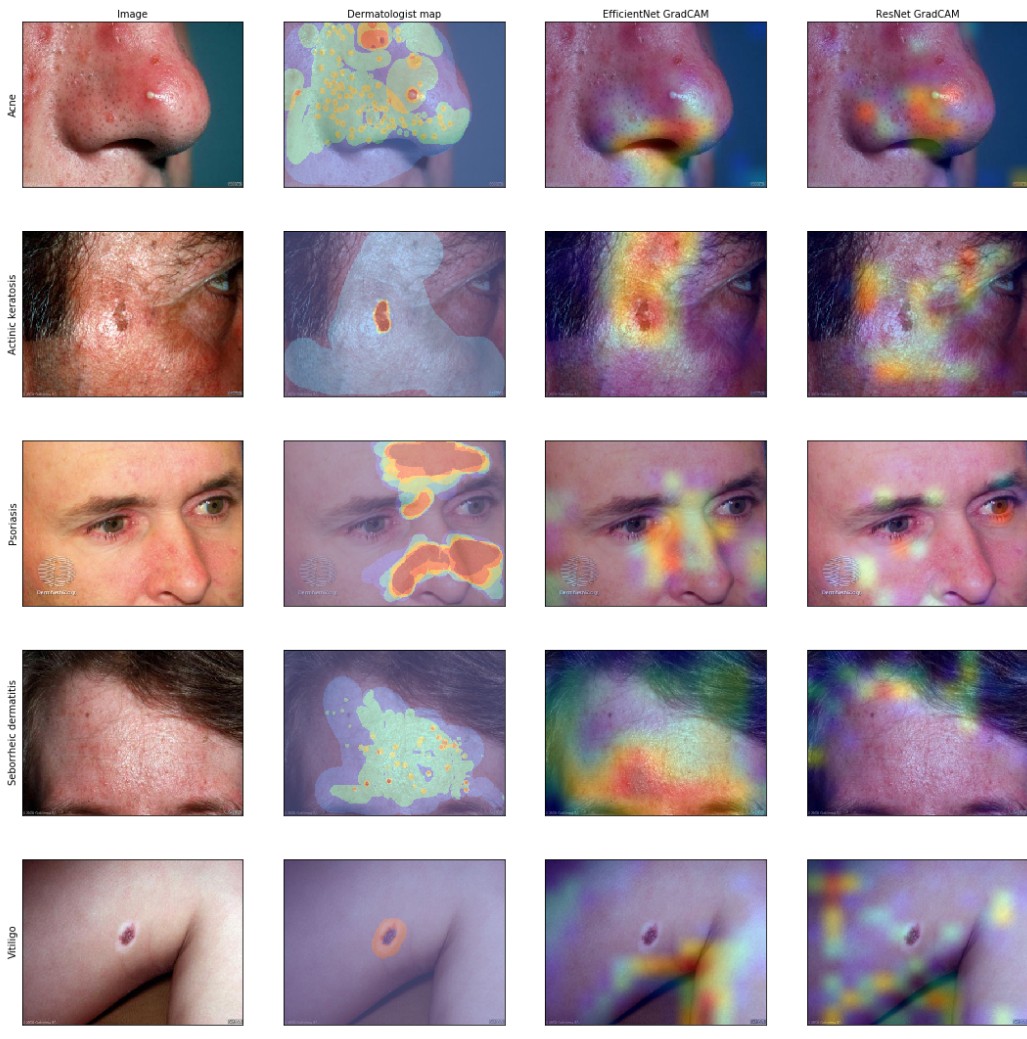

Figure 4: Explanation for images where ResNet correctly predicted the class, while EfficientNet did not. From left to right: the original image, the union of all characteristics selected by all dermatologists labelling the image, an EfficientNetB4 Grad-CAM visualisation, and a ResNet-50 Grad-CAM visualisation.

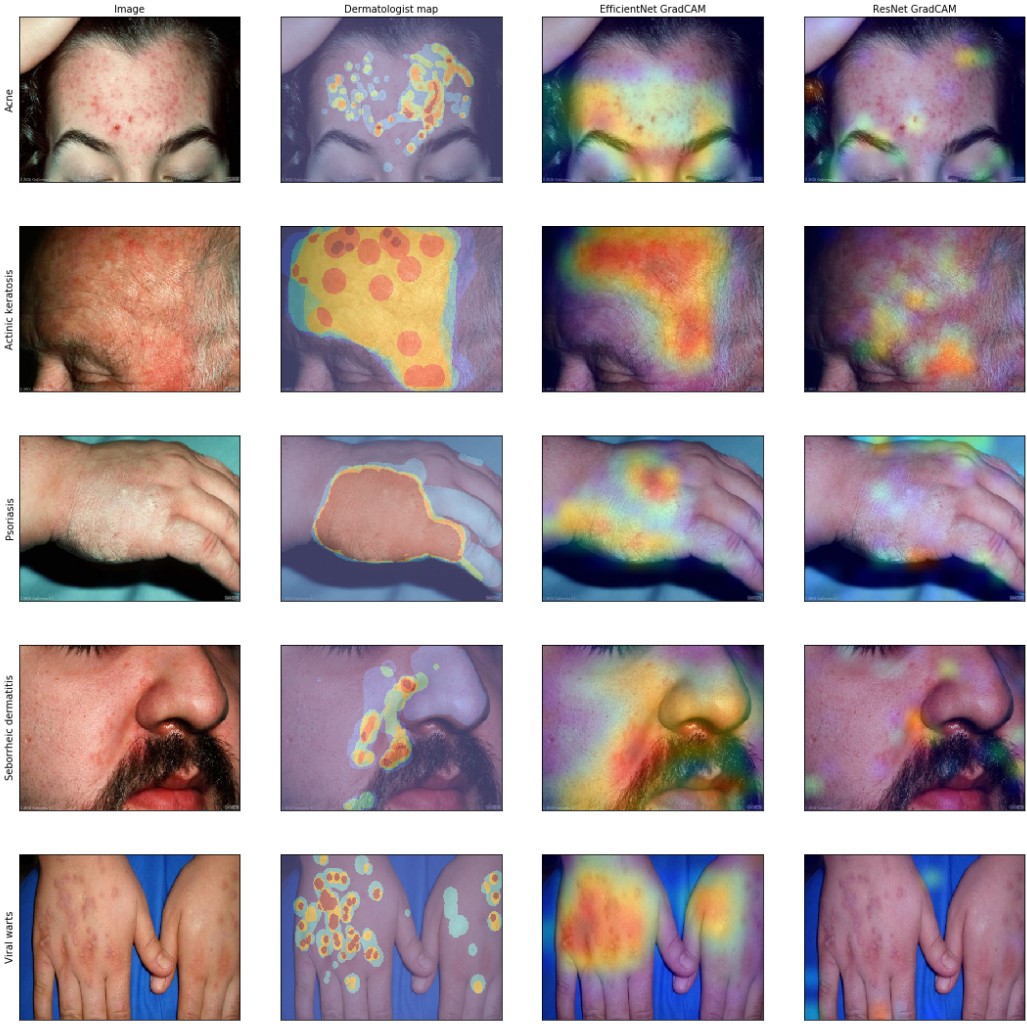

Figure 5: Explanation for images where EfficientNet correctly predicted the class, while ResNet did not. From left to right: the original image, the union of all characteristics selected by all dermatologists labelling the image, an EfficientNet-B4 Grad-CAM visualisation, and a ResNet-50 Grad-CAM visualisation.

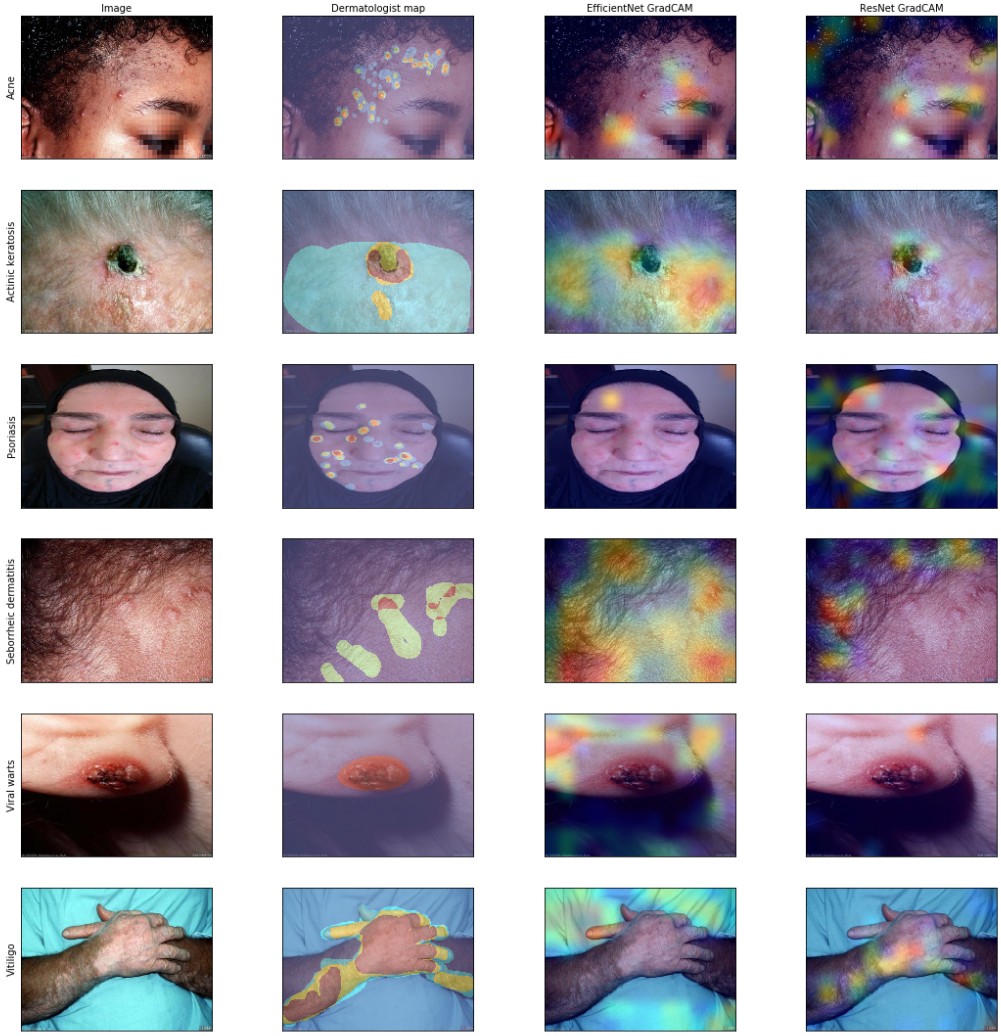

Figure 6: Explanation for images where neither of the two models correctly predicted the class, while EfficientNet did not. From left to right: the original image, the union of all characteristics selected by all dermatologists labelling the image, an EfficientNet-B4 Grad-CAM visualisation, and a ResNet-50 Grad-CAM visualisation.

