# OpenReview forum: "DermX: a Dermatological Diagnosis Explainability Dataset"
_NeurIPS.cc/2021/Track/Datasets_and_Benchmarks/Round1 — Submitted to NeurIPS 2021 Datasets and Benchmarks Track (Round 1)_

### Official Review · Reviewer_NHqH · 2021-06-20
**dataset size is small; benchmarking experiments are not sufficient**

**Rating:** 4
**Confidence:** 4

**Strengths:**

Paper is well-written. The content is clear. Data collection procedures are well-explained. Model implementation details are sufficient.

Attention map using Grad-CAM is one good way for model explainability.

A dermatology dataset with detailed annotations from experts is valuable for machine learning in the medical area.

**Weaknesses:**

1. I am mostly concerned about the dataset size. DermX only contains 525 images. This largely reduces the contribution of the paper.
2. The number of comparative methods is limited. As a baseline, it would be great to add in SVM and a chance model to assess how hard the task is.
3. The proposed model has limitations in diagnosis application. If the ultimate goal is to diagnose diseases on these images, this can be formulated as a binary classification problem (yes: it is this disease or no). The current model proposed by the authors outputs probability for six pre-defined skin disease types. What happens if this is another type of skin disease out of the 6 types it has been trained on. Negative samples (training images out of all six diseases) should be fed into the model as well.
4. In terms of evaluation metrics, authors have been focusing on F1, sensitivity, and specificity. It would be great to include overall classification accuracy in the paper. I think it is also useful to provide intuitive explanations about what these three metrics are emphasizing differently in the paper so that it would help readers interpret the results better.
5. The ways proposed by the authors to assess the model interpretability are limited. In addition to attention maps, if there are non-localizable annotations (technical terms in the form of text words), collected by experts, why not introduce other measures to assess model interpretability? e.g. a GPT3 model outputs key terms for disease diagnosis and it can be used to compare with the terms defined by experts.
6. Based on Fig3, the confusion results show the prediction results are pretty bad (so many off-diagonal predictions). This implies all these two baselines proposed by the authors are not attractive.
7. Both models are trained on 3214 images. Why are the training images not released?

**Additional Feedback:**

Minors:
1. Fig4: the image captions might be wrong. "where ResNet correctly predicted the class, while EfficientNet did not." Based on the attention maps, it seems that EfficientNet is better than ResNet due to its large overlapping areas with human experts
2. Fig5: the image caption is also wrong. "where EfficientNet correctly predicted the class, while EfficientNet did not"


**Clarity:**

Yes, the paper is well-written and clear with enough implementation details for reproducibility.

**Correctness:**

The way the dataset is constructed is sound and it seems correct to me.
The number of evaluation methods is limited. The way of evaluating models and comparing with humans is also limited. (see weakness above)
Based on the confusion matrix, the disease classification results are bad. Given the dataset is very small, there might be an overfitting problem. It is unclear to me, how these models perform in the validation set.

**Documentation:**

Yes, the link for the dataset is provided. The authors also promised to expand the dataset to incorporate more types of skin diseases in the future.

**Ethics:**

I did not examine the source code of the dataset to ensure that the patients' identities have been removed from the dataset due to privacy issues.

**Relation To Prior Work:**

Yes, it is clear that the authors emphasized the differences between DermX and the existing datasets.

**Summary And Contributions:**

The authors introduced DermX dataset containing 525 images in total spanning 6 types of skin disease. For each image, annotations, such as localizable and non-localizable as well as other terms, are provided for disease diagnosis by 8 dermatologists. Two feedforward neural networks are fine-tuned on this task. Attention maps generated using Grad-CAM are compared with segmentation masks from dermatologists as a measure of model explaintability.

---

### Official Review · Reviewer_Taqc · 2021-06-21
**Dermatological dataset and explainable AI**

**Rating:** 4
**Confidence:** 4
**Clarity:** Very well written paper.   No problem…

**Strengths:**

To my knowledge, this dataset is unique and thus fills up a gap.  It could also have important social impacts when considering the possibility for people around the world to picture skin lesions and having it analyzed remotely.   Also, the fact that no less than 8 medical experts manually annotated these images is remarkable.  I would also add that the paper is very well written and that the motivation for such dataset is well argued.

**Weaknesses:**

The paper is not void of interest but it suffers from enough limitations to prevent me from accepting it.

*  As mentioned before, this 525 images dataset can be seen as a major achievement when considering the burden of having 8 medical experts manually segment one-by-one these images.  However, from a pure machine learning perspective, this dataset is too small for its intended purpose.  Indeed, when looking at the F1 scores reported in Table 6, the explainability scores of the 2 CNN are so catastrophically low that one can only conclude that the problem is too difficult for a dataset of this size.  While the authors argue that "results demonstrate that there is still a considerable gap among explanations provided by the models trained for this task and the expert dermatologists." I personally see these results as a demonstration of the limitations of the proposed dataset.

* The paper contains [too] little information on the images.  There is no information on skin color distribution, no mention of any ethical concerns nor image acquisition protocol.

* No intra-expert variability is provided.

* It would have been nice to have further meta information like age, gender, etc.

**Additional Feedback:**

i would be curious to see how a segmentation network (UNet, nnUNet, deepLabV3, etc.) would do on this dataset.

**Correctness:**

The dataset seams to be well balance (c.f. Table 1) and classification + Grad-CAM code is provided.  Unfortunately, there is no centralized evaluation system with a leaderboard that people can relate to.

**Documentation:**

Data is well organized.  However, since the data come from 2 other sources (SD-260 and DermNetNZ) and that the authors were not granted permission to redistribute the original images,  DermX is not self-sufficient and shall always depend on the availability of these other datasets.

**Ethics:**

Patient identity has been removed.  However, I did not see any further information regarding ethics.

**Relation To Prior Work:**

Considering that this dataset is unique, prior work description is not crucial in my opinion.

**Summary And Contributions:**

The proposed dataset contains 525 images of skin lesions spanning 6 classes of diseases.   The dataset has been annotated by 8 medical experts.  Experts were ask not only to label the disease but also to provide supporting explanation for their decision.  The provided explanation is in the form of segmentation maps that subsequent classification CNNs should be capable of recovering.

The authors report an high inter-expert agreement for disease classification (94% and up) except maybe for Seborrhoeic dermatitis and actinic keratosis.  They also report lower inter-expert agreement for supporting characteristics.

The paper reports the classification scores of two CNNs  (namely resNet and efficientNet) and used Grad-CAM to generate attention maps that could serve as supporting characteristics.

---

### Official Review · Reviewer_yeym · 2021-07-04
**Dermatology dataset with segmentations to evaluate explanations. Interesting but needs more work.**

**Rating:** 6
**Confidence:** 3
**Correctness:** All of the claims are correct to the …

**Strengths:**

- This dataset was constructed soundly with a good number of board-certified dermatologists. Additionally, it considers a classic dermatological ontology when deciding which diseases to consider and the localisable characteristics.
- The empirical evaluation of the dermatologists’ evaluations / segmentations was rigorous.
- The effect of distribution shift was also evaluated by training on an internal dataset and testing on DermX which provides realistic results.


**Weaknesses:**

- The major weakness of this paper is the evaluation of the GRAD-CAM explanations with the dermatologist segmentations.
- Currently, a fuzzy implementation of F1, sensitivity, and specificity are used. I believe that dice score is an important metric that is used to compare segmentations to ground-truths.
- I believe that a more rigorous evaluation in the benchmarking for the explanations is needed. A variety of metrics should be used and perhaps even a qualitative evaluation from dermatologists comparing their own segmentations to the generated ones would be helpful


**Additional Feedback:**

- I think that this study is on the right track to creating a benchmark for evaluating explanations in ML for dermatology but it needs more work.
- As stated above, more rigorous evaluations of the explanations would be helpful
- Also more exposition and care is needed with respect to skin tone representation

**Clarity:**

The paper is well-written.


**Documentation:**

Data is readily available and documented on the corresponding Github.


**Ethics:**

One ethical issue I see is the lack of discussion on the distribution of skin tones in the dataset. I noticed one set of images from an individual with darker skin tone but it would be great to understand the distribution of images with respect to skin tone. It’s important that dermatological datasets are representative of a diverse set of skin tones as existing datasets fail at this.


**Relation To Prior Work:**

I think the relation to prior work could be made more clear. For example, how does the dataset size and evaluation of annotations compare to other datasets? Additionally, could training on other dermatological datasets and testing on DermX be possible? If so, I think that would be a great addition. For example, the ISIC dataset comes to mind as a potential candidate to use in training.

**Summary And Contributions:**

This study constructed a new dermatological dataset for the purpose of evaluating the explanations of deep classification models with respect to localizations from board-certified dermatologists. A benchmark is provided on this dataset using two commonly used convolutional architectures.

---

### Decision · Program_Chairs · 2021-07-26

**Decision:**

Reject

**Comment:**

A few concerns are raised by reviewers on this submission. First and foremost, a few ethical concerns were raised. Reviewers pointed out that there is no statement regarding the ethical concerns around darker skin images in the dataset. We find authors’ responses unsatisfactory—although they conducted some statistical analysis upon this review, they argue that they sub-selected images from a publicly available dataset, and therefore, refer to their ethical statements instead of their own. We respectfully disagree—a new dataset and re-distribution of an existing dataset carry equal responsibility; folks involved in re-distribution have to own any benefits/issues with the original dataset. We strongly recommend authors consider the ethical concerns of this dataset as if it is their own, and address this accordingly. Second of all, reviewers point out that the  number of datapoints is too small (525 images). One questions why not also releasing 3000+ more images used for training (as described in the paper), to which authors responded that those images could not be released due to IP issues. Authors argued that they expect users to use this as a validation set, rather than training a new model. But doing so introduces a new problem of potential distribution shifts. An additional discussion on this would be useful in future submissions of this work. Third, a few reviewers also pointed out that they wanted to see more rigorous evaluations.